# Curcumin and Weight Loss: Does It Work?

**DOI:** 10.3390/ijms23020639

**Published:** 2022-01-07

**Authors:** Kamila Kasprzak-Drozd, Tomasz Oniszczuk, Marek Gancarz, Adrianna Kondracka, Robert Rusinek, Anna Oniszczuk

**Affiliations:** 1Department of Inorganic Chemistry, Medical University of Lublin, Chodźki 4a, 20-093 Lublin, Poland; kamilakasprzakdrozd@umlub.pl; 2Department of Thermal Technology and Food Process Engineering, University of Life Sciences in Lublin, Głęboka 31, 20-612 Lublin, Poland; 3Institute of Agrophysics, Polish Academy of Sciences, Doświadczalna 4, 20-290 Lublin, Poland; m.gancarz@ipan.lublin.pl (M.G.); r.rusinek@ipan.lublin.pl (R.R.); 4Faculty of Production and Power Engineering, University of Agriculture in Kraków, Balicka 116B, 30-149 Kraków, Poland; 5Department of Obstetrics and Pathology of Pregnancy, Medical University of Lublin, 20-081 Lublin, Poland; adriannnakondracka@umlub.pl

**Keywords:** curcumin, natural compounds, obesity, adipogenesis, lipid metabolism

## Abstract

Obesity is a global health problem needing urgent research. Synthetic anti-obesity drugs show side effects and variable effectiveness. Thus, there is a tendency to use natural compounds for the management of obesity. There is a considerable body of knowledge, supported by rigorous experimental data, that natural polyphenols, including curcumin, can be an effective and safer alternative for managing obesity. Curcumin is a is an important compound present in *Curcuma longa* L. rhizome. It is a lipophilic molecule that rapidly permeates cell membrane. Curcumin has been used as a pharmacological traditional medicinal agent in Ayurvedic medicine for ∼6000 years. This plant metabolite doubtless effectiveness has been reported through increasingly detailed in vitro, in vivo and clinical trials. Regarding its biological effects, multiple health-promoting, disease-preventing and even treatment attributes have been remarkably highlighted. This review documents the status of research on anti-obesity mechanisms and evaluates the effectiveness of curcumin for management of obesity. It summarizes different mechanisms of anti-obesity action, associated with the enzymes, energy expenditure, adipocyte differentiation, lipid metabolism, gut microbiota and anti-inflammatory potential of curcumin. However, there is still a need for systematic and targeted clinical studies before curcumin can be used as the mainstream therapy for managing obesity.

## 1. Introduction

Currently, the numbers of people classified as either being overweight or obese constitute a serious global public health problem. According to 2016 data, over 39% (1.9 billion) of the world’s adults were overweight, while over 13% were obese [1]. It is estimated that by 2030, nearly 60% of the world’s adult population, approximately 3.3 billion people, will be overweight or obese [2]. Obesity is a negative factor, as it contributes to increased morbidity and mortality. The World Health Organization (WHO) has classified obesity as a chronic non-infectious disease, in addition to diseases such as cardiovascular diseases, diabetes, cancer and some diseases of the gastrointestinal tract. In most developed and developing countries, these diseases are the leading cause of death, reduction of life expectancy and deterioration of it, as well as the social costs associated with their treatment [3]. 

An important parameter describing this disease is Body Mass Index (BMI). BMI is calculated by dividing weight in kilograms by height in meters squared. Overweight in adults is defined as a BMI of higher than or equal 25 and obesity as equal or more than 30 [4,5]. Numerous epidemiological studies have shown that the risk of death increases with increasing BMI. Obesity-related mortality results mainly from the much higher incidence of cardiovascular diseases: arterial hypertension, heart failure, ischemic heart disease, pulmonary heart disease, pulmonary embolism, strokes and varicose veins of the lower extremities. Other complications associated with obesity that are important for health and life are diabetes and cancer [6,7]. Metabolic complications are associated especially with abdominal obesity [8,9]. According to WHO, overweight and obesity are responsible for 80% of all type 2 diabetes, 35% of all ischemic heart disease and 55% of all hypertension cases. Treating the associated health risks and related complications is as important as managing obesity [10]. Clinical studies show that weight reduction brings significant health benefits, including improvement of metabolic indicators such as lipid disorders, carbohydrate metabolism disorders or increased blood pressure [6]. 

The basis of obesity treatment is always a reduced calorie diet and increased exercise. In justified cases, pharmacotherapy is used, but it is associated with possible side effects. Currently used drugs include orlistat, lorcaserin, phentermine/topiramate and naltrexone/bupropion. Diet has been known for many years to play a key role as a risk factor for chronic diseases [3]. Hence, there are continuous efforts being made to discover and evaluate natural compounds with minimal side effects for weight management [1]. One such substance is curcumin (1,7-bis(4-hydroksy-3-metoxyphenyl)-1,6-heptadien-3,5-dione)—the polyphenolic active ingredient of the spice turmeric (*Curcuma longa* L.)—which belongs to the ginger family (*Zingiberaceae*) [11,12]. Curcumin is a polyphenolic compound that represents the most important curcuminoid isolated from the rhizome of the plants [13]. It is believed that curcumin possess subtle effects through multiple mechanisms and biochemical targets, collectively leading to substantial health benefits. This indicates that this compound has potential for preventing (and, in the future, treating) obesity [1,12]. In this context, this review documents and evaluates the anti-obesity potential of curcumin with a special focus on its mechanisms of action in regulating obesity. The evidence of anti-obesity functions of this polyphenol is drawn from published studies that have used both in vitro and in vivo experimental methods.

Polyphenols are plant secondary metabolites that have an aromatic ring with at least one hydroxyl group. Their structure can vary from simple compounds, to complex polymers with high molecular weight. Polyphenols may be subdivided into two main groups: flavonoids (i.e., anthocyanins, flavanols and flavanones) and non-flavonoids (i.e., phenolic acids, xanthones, stilbens and tannins) [14]. Polyphenols have antioxidant properties, and this is due to the phenolic hydroxyl structure, in which the electrons have a conjugation effect. Thus, the ability to bind hydrogen ions is weakened. It becomes more prone to dissociation, so the active hydrogen ion neutralizes free radicals and other reactive oxygen species, hence scavenging free radicals [15].

## 2. Medicinal Activities, Bioavailability and Metabolism of Curcumin—Brief Overview

*Curcuma longa* has been used for centuries in traditional Asian medicine, and nowadays, it is widely adopted in international cuisine as a dietary spice [16]. Curcumin has many properties, including antioxidant. The role of antioxidants during the occurrence of oxidative stress is important, which in turn may contribute to the prevention or delay of the development of many diseases (including civilizational) and their possible complications. Therefore, naturally derived antioxidants such as curcumin are of high clinical value. The anti-inflammatory effect of curcumin is equally significant. Curcumin inhibits and regulates tissue production and secretions of pro-inflammatory cytokine, such as interleukins or tumor necrosis factor alpha (TNF-α) [17]. 

Curcumin is a potential anti-cancer agent because of its multidirectional properties with regard to the signaling/molecular pathways. Curcumin possesses the ability to modulate the core pathways involved in cancer cell proliferation, apoptosis, cell cycle arrest, paraptosis, autophagy, oxidative stress and tumor cell motility [18]. 

Curcumin and *Curcuma longa* extract inhibit the growth of microorganisms, both G(+) and G(-) bacteria, which often cause human infectious diseases. Curcumin is known to have an antiviral effect [17]; it also has been suggested as a potential treatment option for patients with COVID-19 [19]. Curcumin has an anti-diabetic effect through, for example, enhancing glucose uptake and improving pancreatic beta cell function. In addition, curcumin contributes to the reduction of the gluconeogenesis process and to the increase of glucokinase activity [20]. Curcumin was able to restore oxidative stress and DNA methyltransferase functions against diabetic retinopathy [18]. Moreover, curcumin has shown anti-lipidemic effects. There are animal model studies on the use of curcumin in pregnancy. Due to the anti-inflammatory activity of this compound, the possibility of using this compound in the treatment of complications of pregnancy has been suggested, including Gestational Diabetes Mellitus, PreTerm Birth, Preeclampsia and exposure to toxic agents and pathogens [21].

Other activities of curcumin worth mentioning include immune modulation, cardiovascular protection, anti-pulmonary fibrosis, anti-chronic obstructive pulmonary disease and anti-dementia activity [17]. There is a growing scientific interest in curcumin’s therapeutic potential. Indeed, more and more clinical trials based on curcumin administration have been published or are underway [13].

The systemic bioavailability of curcumin is low, possibly due, at least in part, to metabolism [22]. In addition, it is absorbed from the gastrointestinal tract to a minor extent. This is mainly due to water insolubility, rapid metabolism and excretion [23]. In one study, after oral administration of curcumin at a dose of 0.1 g/kg to mice, the peak plasma concentration of free curcumin was only 2.25 μg/mL [24]. This study involved participants who consumed 3.6 g of curcumin per day. Semiquantitation showed a curcumin plasma concentration of only 11.1 nmol/L [25]. Furthermore, when curcumin was consumed in doses as high as 8 g per day, plasma levels of free curcumin were also negligible. Circulating curcumin was, however, detectable as drug in glucuronide and sulfate conjugate forms [26]. Other human studies have shown that the maximum concentration (before four hours) of 10 g of curcumin administered orally is 50.5 ng/mL, while that of 12 g is 51.2 ng/mL [27]. Importantly, the bioavailability of curcumin can be influenced by the food matrix (e.g., lipids and proteins) [16].

Curcumin metabolites exhibit the same physiological and pharmacological properties, and also have similar properties and potency [16,23]. Curcumin is metabolized mainly in the liver, together with the intestine and gut microbiota (GM) [28]. Enzymes of the large intestine metabolize curcumin and this occurs in two phases (Figure 1). [23]. Phase-I metabolism involves the reduction of double bonds by which curcumin is converted into di, tetra, hexa and octahydrocurcumin (IUPAC names being, respectively: 1,7-bis(4-hydroxy-3-methoxyphenyl)hept-1-ene-3,5-dione, 1,7-bis(4-hydroxy-3-methoxyphenyl)heptane-3,5-dione, 5-hydroxy-1,7-bis(4-hydroxy-3-methoxyphenyl)-3-heptanone, 1,7-bis(4-hydroxy-3-methoxyphenyl)heptane-3,5-diol). 

Subsequently, in phase-II metabolism, glucuronide or sulfate is conjugated to the curcumin and to its hydrogenated metabolites [23,29]. 

According to Ireson et al. [22], curcumin undergoes extensive metabolic conjugation and reduction in the gastrointestinal tract, and there is a more considerable metabolism in human intestinal tissue than in rat intestinal tissue. An in vitro study showed that curcumin glucuronide is the most frequently detected metabolite after incubation of curcumin with the microsomal fraction of rat and human intestinal and liver tissue homogenates [22]. In an animal model (rats), after oral administration of curcumin, mainly its metabolites as conjugates (curcumin glucuronide and curcumin sulfate) were detected in the blood. However, in free form, such content is insignificant. Likewise, glucuronide and sulfate conjugates are detected in humans after oral administration of curcumin, but as a free form, it is barely detectable [30].

Transformation of curcumin also takes place through the interaction of enzymes produced by GM residing in the colon. Curcumin intestinal transformations include several steps and different classes of microbial enzymes. For this reason, the composition of the gut microflora will result in differences in the biotransformation of curcumin in the daily diet. Several enteric bacteria capable of modifying curcumin have been identified [13]. An alternative metabolism of curcumin occurs by intestinal microbiota of commensal *Escherichia coli*. The enzyme that converts curcumin (CurA produced by bacteria) induces a two-step reduction: curcumin being converted NADPH-dependently into an intermediate product, dihydrocurcumin, and then the end product, tetrahydrocurcumin [31]. Tan et al. [32] identified the other bacteria that are characterized by the ability to modify curcumin. These are *Escherichia fergusonii* (ATCC 35469) and two strains of *E. coli* (ATCC 8739, DH10B). They produce dihydrocurcumin, tetrahydrocurcumin and ferulic acid [32]. Other microorganisms, such as *Bifidobacteria longum*, *Bifidobacteria pseudocatenulaum*, *Enterococcus faecalis*, *Lactobacillus acidophilus*, and *Lactobacillus casei* [13], *Blautia* sp. *Pichia anomalia* and *Bacillus megateriumdcmb*, are all biologically relevant bacterial strains capable of degrading curcumin [23]. 

## 3. Reciprocal Interaction between Curcumin and Gut Microbiota (GM)

There is a paradox between curcumin’s low systemic bioavailability and its broad pharmacological activity. The research hypothesis of this fact is that curcumin directly exerts a regulatory effect on the intestinal microflora. The composition of the human GM is influenced by many environmental and lifestyle factors. Therefore, any disturbance in the gut microbiome profile or dysbiosis may play a key role in disease progression in humans. Moreover, curcumin and its metabolites influence the microbiota. There are two different phenomena in the reciprocal interaction between curcumin and GM: the regulation of the intestinal microflora by curcumin and the biotransformation of curcumin by the intestinal microbiota. Importantly, both are potentially crucial for curcumin’s activity [16]. Bidirectional interaction between curcumin and gut microbiota are shown in Figure 2 [16]. 

### Curcumin, Gut Microbiota and Obesity

The primary function of the gut microbiota is to contribute to the digestion of fiber and other nutrients. Due to microbiota, 4–10% of the daily caloric value is obtained (80–200 kcal/day) [33]. Furthermore, GM are recognized to play a key role in maintaining immune and metabolic homeostasis and protection against pathogens [34]. Disorders of the GM can result in numerous consequences that adversely affect the metabolism, immune and endocrine systems, energy homeostasis of the organism and lipid metabolism. In addition, they lead to the disturbance of the proper structure and function of the intestinal barrier, the damage to which may result in the penetration of microorganisms or their fragments (LPS—lipopolysaccharide) into the body and cause chronic inflammation [35]. As a consequence, many metabolic pathways are activated, which may lead to increased energy extraction from food (e.g., activation of the G-PCR protein-coupled receptor and increased production of short chain fatty acids), increased fat storage (e.g., decreased lipoprotein lipase—fasting-induced adipose factor FIAF [33,35] and activation of the endocannabinoid system [36]), impairment of its utilization in muscles (e.g., inhibition of the activity of protein kinase activated by AMP-activated protein kinase, AMP-AMPK) [33,35] and a hormonal mechanism consisting in suppresses insulin-mediated fat accumulation via the short-chain fatty acid receptor GPR43 [37], among others. It is worth underlining that it is still unclear how these mechanisms interact to influence the overall metabolic status of an individual [35].

The effect of probiotics in counteracting obesity is strain-specific. Various in vitro and in vivo studies have shown that probiotics, especially *L. casei*, *L. rhamnosus*, *L. gasseri*, *L. plamarum* and *Bifidobacterium* (*B. longum*, *B. breve*, *B. animalis*), could potentially reduce weight gain and improve some of the associated metabolic parameters. The latter may become an effective strategy for the prevention and treatment of obesity in adult individuals [33,38]. The mucinogenic G (-) bacterium *Akkermansia municiphila* constitutes 3–5% of the microbial community in healthy people and its amount in the gastrointestinal tract is inversely correlated with body weight, both in humans and in experimental animals [33]. Studies have been published demonstrating that polyphenolic compounds may affect the diversity of the gut microbiome [39,40], and increase the presence of *Akkermansia muciniphila* [39].

After oral administration (when it is expected to be present in high concentrations in the gastrointestinal tract), curcumin is distributed throughout the intestines, whereupon it affects GM microbial richness, diversity and composition [23]. Based on studies in an animal model (mouse), it was found that that curcumin administration exerts significant effects on GM family members, such as the *Bacteroidaceae*, *Rikenellaceae* and *Prevotellaceae* [41]. One of the studies noted that curcumin also improves the composition of gut microbiota in colitis mice. Herein, the addition of curcumin ameliorated dextran sulfate sodium to the diet of mice in the case of colitis induced by dextran sulfate sodium, resulting in a smaller loss of the relative abundance of Akkermansia, compared to a test without the curcumin addition [42].

Curcumin has been demonstrated to have a weight-loss effect in a menopausal rat model induced by ovariectomy. This compound (administered 100 mg/kg per day) promoted GM, including *Anaerotruncus*, *Exiguobacterium*, *Helicobacter*, *Papillibacter*, *Pseudomonas*, *Serratia* and *Shewanella*. An estrogen deficiency caused by ovariectomy induced alterations in the distribution and structure of the intestinal microflora in rats, and the administration of curcumin was able to partially reverse the changes in the diversity of the GM [43].

Gut microbial homeostasis and gut barrier integrity are impaired in high-fat diet-induced obesity (HFD), where increased numbers of gram-negative bacteria, mainly Bacteroides spp., a major source of LPS, are evidenced [2]. In obesity, the bacterial LPS penetrates the damaged intestinal wall and reaches the tissues through the systemic circulation [44]. In obese white adipose tissue, LPS activates pro-inflammatory signaling, and consequently, e.g., interlukin-6 and monocyte chemoattractant protein-1, and this may be the cause of metabolic dysfunction. In a very recent study conducted by Islam et al., the links between supplementing high-fat diet (HFD) with curcumin, adipose tissue inflammation and changes in gut microbiota were investigated. The administered dose to obese mice was consistent with a human equivalent dose of 2 g per a day (calculated per 60 kg adult human) over a period of 14 weeks. Briefly, therein, curcumin supplementation significantly reduces adiposity and total macrophage infiltration in white adipose tissue, compared to HFD group. 

According to the results, the relative abundance of the *Lactococcus*, *Parasutterella* and *Turicibacter* genera are increased in the HFD supplemented with curcumin, as compared with HFD without. It is believed that curcumin exerts protective metabolic effects in dietary obesity, in part through downregulation of adipose tissue inflammation, which may be mediated by alterations in the composition of gut microbiota, and the metabolism of curcumin into curcumin-O-glucuronide [2].

Wang et al. [45], as a consequence of their research, stated that curcumin attenuates the high-fat high-cholesterol found in Western-type diet (WD)-induced chronic inflammation and associated metabolic diseases (including obesity). This comes about by modulating the function of intestinal epithelial cells (IEC) and the intestinal barrier function. Human IEC lines were used for their studies. In the study, the modulation of direct and indirect effects of LPS on intracellular signaling, as well as tight junctions, was examined. The outcome of the work was the recognition that pretreatment with curcumin significantly attenuated LPS-induced secretion of master cytokine IL-1β from IECs and macrophages. Furthermore, curcumin also reduced IL-1β-induced activation of p38 MAPK in IECs. It is, therefore, considered that the IEC and the intestinal barrier are possibly the main sites of action of curcumin. Furthermore, curcumin, by reducing the dysfunction of the intestinal barrier, modulates chronic inflammatory diseases despite its low bioavailability [45].

Curcumin has been shown to increase energy expenditure. Growing evidence points to a strong link between the GM and energy metabolism. It is estimated that up to a third of the metabolites found in the blood of mammals are derived from the GM. Bile acids (BA) belong to the class of such metabolites. They are produced in the liver and biochemically modified by intestinal bacteria. BA can act as signaling molecules in many tissues and influence host metabolism through signaling pathways through BA detecting receptors [46]. Among them, BAs have been found to increase the activity of BAT via G protein-coupled bile acid receptor 5 (TGR5) [47]. Activation of TGR5 induces mitochondrial disconnection and oxygen consumption by upregulating cyclic adenosine monophosphate (cAMP) production and the cAMP/protein kinase A (PKA) signaling pathway [48]. 

Han et al. [46] investigated whether GM mediates the effects of curcumin in improving energy homeostasis. For this purpose, mice fed a high-fat diet (HFD) uncoupling protein 1 (Ucp1) knockout and G protein-coupled membrane receptor 5 (TGR5) were used. The dose of curcumin administered is 100 mg/kg^−1^. It was found that mice fed an HFD concomitantly treated with curcumin showed reduced body weight gain and increased cold tolerance due to increased adaptive thermogenesis compared to control mice. In addition, the anti-obesity effects of curcumin were abolished by the Ucp1 knockout. Further work involving 16 S ribosomal DNA sequencing analysis showed that curcumin restructured the GM in these mice. The administration of curcumin changed the metabolism of BA—the content of deoxycholic acid (DCA) and lithocholic acid (LCA) (the two strongest ligands for TGR5) increased. The study revealed that in order to activate the cAMP/PKA signaling pathway in thermogenic adipose tissue through curcumin, interaction of GM and TGR5 is needed [46].

Investigations of the connection between circulating bacterial endotoxin lipopolysaccharide (LPS) and metabolic diseases has shifted the focus from actual bacterial infections in the etiology of these diseases, to increased translocation of bacterial products (e.g., LPS) due to rise in intestinal permeability. Published research suggest that oral supplementation with antibiotics (100 mg/L Neomycin and 10 mg/L Polymyxin B per day) or curcumin (100 mg/kg per day) significantly improve intestinal barrier function and attenuated the release of LPS, yet contemporaneously did not affect the food intake or Western diet-induced weight gain [49].

Obesity is the accumulation of abnormal or excessive fat that may interfere with the maintenance of an optimal state of health, including excessive release of inflammatory mediators [50]. The reason for the positive effect of the turmeric extract on the reduction of the expression of inflammatory cytokines in adipose tissue was investigated. For this purpose, mice that were fed either a control diet, a high-fat (HF) diet or an HF diet containing *Curcuma longa* extract (0.1% of curcumin in the HF diet) associated with white pepper (0.01%) were used. It was pointed out that this effect may be more related to the direct influence of bioactive metabolites reaching the adipose tissue than from changes in the gut microbiota composition. Administration of the preparation containing turmeric did not, however, change the total bacterial content in the cecum (determined after 4 weeks of the HF diet). Thus, it had no effect on the major gram-positive species capable of controlling systemic inflammation (*Bifidobacterium* spp. and *Lactobacillus* spp.), nor on the major G(-) bacteria (*Bacteroides/prevotella* spp.), which can release pro-inflammatory LPS [51]. 

There is a lot of research into the effects of turmeric on the organism. However, there are still a small number of articles that unequivocally explain the GM-induced curcumin metabolism, its action over intestinal permeability, and effect on obesity and/or inflammation [52].

## 4. Redox Balance in Obesity, ROS and Inflammation in Obesity

Oxidative stress plays significant role in the pathophysiology of obesity, increasing inflammatory response in adipose tissue, stimulating differentiation of preadipocytes to mature adipocytes, suppressing fatty acids (FAs) oxidation and promoting lipogenesis.

The antioxidant activities of curcumin and its therapeutic potential in obesity prevention (and treatment) depend on its activities as reactive oxygen species (ROS) scavengers and on its capacity to prevent the activation of NF-κB (nuclear factor-κB), and reduce the expression of target genes, including those participating in inflammation [53].

Intracellular ROS are mainly produced in the mitochondria. ROS may also be produced by the endoplasmic reticulum (ER), lysosomes and peroxisomes, as well as by cytosolic enzymes. ROS have many positive biological effects at low concentrations, e.g., the defense against pathogenic microorganisms. Nevertheless, at high levels, they may damage DNA, proteins and lipids, resulting in tissue and cell destruction [54,55]. To maintain ROS at appropriate levels, tissues have antioxidant compounds (glutathione, ubiquinone, thioredoxin and urate) that work synergically. The human body also has proteins (transferrin, ferritin, caeruloplasmin and lactoferrin), with antioxidant properties and antioxidant enzymes (glutathione peroxidase (GPx), glutathione reductase, glutathione S-transferase (GST), superoxide dismutase (SOD), catalase, peroxiredoxins (Prx), thioredoxin reductase, and NAD(P)H:quinone oxidoreductase 1 (NQO1)) [54,55]. In addition, novel types of antioxidant enzymes (*paraoxonase* spp.) that participate in diseases linked to obesity have been discovered [56].

Phagocytic leukocytes, i.e., eosinophils, neutrophils, monocytes and macrophages, contribute to the production of oxidizing substances which attack tissues. ROS induce the inflammation by activating NF-κB and overproduction of cytokine [57]. A systemic inflammation is caused by the increased number and size of adipocytes. These cells secrete adipocytokines, which include anti-inflammatory agents, e.g., adiponectin, transforming growth fac-tor beta (TGFβ), IL-4, IL-10 and IL-13, IL-1 receptor antagonist and apelin, as well as pro-inflammatory agents, e.g., tumor necrosis factor-α (TNF-α), leptin, IL-6, resistin, visfat-in and plasminogen activator inhibitor) [58,59,60]

The NF-κB signaling pathway controls also the activity of inflammasomes [61]. Over-eating affects NF-κB signaling pathways in adipose tissue by interfering with mitochondrial function, which leads to the overproduction of ROS [62]. H_2_O_2_, through tyrosine phosphorylation, causes the degradation of the nuclear factor-enhancing kappa light chains of activated B cells (IκBκ), an NF-κB inhibitor [63]. 

Numerous pro-inflammatory enzymes, e.g., cyclooxygenase-2, arachidonate 5-lipoksygenase and 12-lipoxygenase, are activated after DNA binding. This phenomenon intensifies ROS production and potentiates ROS-induced harm [64]. On the other hand, free fatty acids (FFAs) activate the Toll-like receptor (TLR) pathway, which regulates the expression of numerous inflammatory factors. For example, the TLR2 and TLR4 pathways play significant roles in vascular dysfunction and insulin resistance [65,66]. Excess of ROS upregulates the expression of TLR2 and TLR 4 [67]. Some proposed mechanisms by which FFAs work by binding to TLRs are the formation of ceramides, the stimulation of the activity of several serine/threonine kinases and the increase in the production of free radicals. Probably, these pathways act synergistically [68].

## 5. The Inhibition of Adipogenesis

Adipocytes are the main cellular constituents of adipose tissue. They play a vital role in lipid homeostasis and energy balance. White adipocytes (white adipose tissue—WAT) and brown adipocytes (brown adipose tissue—BAT) are the main mature adipocytes [1]. WAT has a single fat droplet and few mitochondria. This tissue produces hormones that regulate nutrient homeostasis and participate in store fat and in the regulation of food intake by secreting hormones and promoting inflammation. Therefore, WAT plays a key role in obesity. BAT has multiple fat droplets and many mitochondria. This tissue can be activated to oxidize fatty acids to maintain body temperature, and BAT regulates nutrient homeostasis and energy expenditure under specific conditions of physical activity and energy intake. Therefore, BAT may slow obesity. Under exposure to cold or due to activation of β-adrenergic receptors, a transformation of white to beige adipocytes occurs. This is deemed “browning”. Beige adipocytes burn nutrients, and are involved in nutrient homeostasis and adaptive thermogenesis [69]. 

Adipocytes are derived through the process of differentiation of mesenchymal cells into preadipocytes and then to mature adipocytes. Over-normative adipogenesis causes the accumulation of an excessive numbers of adipocytes, leading to obesity [70]. To manage obesity, any of several stages of adipogenesis can be interfered with by a number of transcriptional factors [71]. Some of the important transcription factors involved in adipocyte differentiation belong to a group of CCAAT/enhancer binding proteins (C/EBP), sterol regulatory element binding proteins (SREBP) and peroxisome proliferator-activated receptors (PPAR) [72]. Adipose tissue-derived fatty acid synthase (FAS) can play an important role in adipocyte differentiation; therefore, inhibition of FAS can cause a reduction in adipose tissue [73].

Curcumin is capable of interfering with or constraining several stages of adipocyte differentiation [74]. This compound can alter the life cycle of adipocytes through the suppression of preadipocyte proliferation and mitogenesis, inhibition of adipogenesis and induction of mature adipocyte apoptosis.

Adipogenesis is inhibited by the canonical Wnt signaling cascade, in which β-catenin/TCF functions as the key effector [75,76]. Kim et al. [77] demonstrated that curcumin possesses an anti-adipogenic function, both in the 3T3-L1 murine cell model and in human primary preadipocytes (determined by the intracellular lipid accumulation assay). The same researchers found that this compound impeded mitotic clonal expansion during the early stage of adipocyte differentiation. This process involved the inhibition of genes that encode the early adipogenic transcription factors, e.g., Krüppel-like factor 5, C/EBPα and PPARγ [77]. In 3T3-L1 cells, Ahn et al. [78] found that curcumin decreased adipocyte protein 2 (aP2, a mature adipocyte marker) mRNA expression, increased c-Myc and cyclin D1 expression (Wnt targets) and constrained mitogen-activated protein kinase (MAPK) phosphorylation, which has been associated with 3T3-L1 differentiation into adipocytes. Moreover, during the differentiation process, intake of curcumin enhanced nuclear translocation of β-catenin. These findings show that the Wnt signaling pathway participates in curcumin-induced suppression of adipogenesis in 3T3-L1 cells. Thus, curcumin can suppress 3T3-L1 adipogenesis via stimulating the canonical Wnt signaling cascade [78]. 

Wang et al. [79] conducted further mechanistic exploration with the 3T3-L1 cell model. They found that another Wnt pathway effector molecule, transcription factor 7-like 2 (Tcf7l2), is the target of the microRNA miR-17-5p (a member of the miR-17/92 cluster, which was shown to accelerate 3T3-L1 adipogenic differentiation). Tian et al. found that curcumin treatment attenuated miR-17-5p expression and stimulated Tcf7l2 expression in the 3T3-L1 cell model [80]. Unfortunately, it is still unknown whether curcumin intervention also represses adipogenesis in vivo. It should also be determined whether the in vivo repression plays a role in reducing body weight gain [79].

## 6. Regulation of Lipid Metabolism

Lipid metabolism is a multi-step process involving the synthesis, storage and degradation of fatty acids, triglycerides and cholesterol. Many compounds, for example enzymes and hormones, participate in this action. The level of cholesterol is regulated by physiological synthesis, transportation, absorption and excretion [81]. Acetyl-coenzyme A (acetyl-CoA) is the precursor of cholesterol. Synthesis of acetyl-CoA is regulated by SREBP1a, SREBP2, 3-hydroxy-3-methylglutaryl coenzyme A reductase and low-density lipoprotein (LDL) receptor [82]. Activation of SREBP1c is caused by stress in the endoplasmic reticulum. This process improves transcription of lipogenic enzyme genes [83]. 5’-AMP-activated protein kinase (AMPK) is a valid regulator of the lipid synthesis pathways. AMPK can reduce the synthesis of FAs by interfering with SREBP-1c and FAS. Activation of AMPK causes the oxidation of FAs in the liver and inhibits the synthesis of cholesterol [84]. Peroxisome proliferator-activated receptor α (PPARα) can also have a positive influence on FAs oxidation and cholesterol breakdown in the lipid metabolism process [85]. Modulation of mentioned factors can have desirable effects in obesity prophylaxis.

Curcumin causes AMPK and PPARα activation. The mentioned factors, in turn, cause the inhibition of acetyl-CoA carboxylase (ACC). These processes decrease lipid accumulation by reduction in FAs synthesis and increase in fatty acid oxidation.

In vitro, in vivo studies and clinical trials have shown that plasma lipid elevation induced by a high-fat diet (HFD) can be greatly reduced by the taking orally curcumin [86,87]. 

Tian et al., 2015 [88] assessed the effect of curcumin intake (6-day) in a dexamethasone induced insulin resistance mouse model. The researchers wanted to confirm the hypothesis for the existence of the anti-inflammation-independent insulin sensitizing effect of curcumin. They observed an improvement on insulin tolerance, which was associated with an increase in the expression in the liver of the metabolic hormone fibroblast growth factor 21 (FGF21). FGF21 is mainly produced in the liver and released during fasting [89,90]. Its induction occurs via the activation of the nuclear receptor PPARα [91,92]. During the adaptive starvation response period, the PPARα/FGF21/PGC-1α (peroxisome proliferator-activated receptor gamma coactivator 1-alpha) axis promotes FAs oxidation, gluco-neogenesis as well as tricarboxylic acid cycle flux.

Various types of studies (preclinical and clinical) have documented the insulin-sensitizing effect of FGF21 [93,94,95,96]. It was concluded that obesity represents a state of resistance to FGF21, as increased serum levels of FGF21 have been reported in experimental animals (mice) and in humans [95,97,98]. Furthermore, in C57BL/6 mice fed a low-fat diet (LFD) or in murine or human hepatocytes, it has been noticed that in vitro curcumin therapy has effects such as the stimulation of Fgf21 mRNA expression, production of the hormone FGF21 in the liver and increased levels of FGF21 in plasma. Nevertheless, when curcumin was administered to HFD-fed C57BL/6 mice for 12 weeks, it attenuated the HFD-induced increase in FGF21 in the liver and blood plasma. This effect was due to the partial restoration of the expression in the liver of the genes encoding the FGF21 FGFR1 receptor and the βKlotho coreceptor, as well as the expression of lipolysis genes [89].

Additionally, hepatocytes in mice on HFD for 4 weeks showed attenuated response to ex vivo recombinant FGF21 treatment that was neutralized by administering curcumin to the animals. On this account, it has been concluded that curcumin intervention may improve FGF21 sensitivity in HFD-fed mice. [89]. It has not been established whether the insulin signaling sensitization of curcumin is secondary to the restoration or maintenance of sensitivity to FGF21.

Experiments involving curcumin dietary intervention in animals with hyperlipidemia (induced by a high glucose diet) resulted in a reduction in the blood of free total cholesterol, fatty acids and triglycerides (where the effect was particularly evident) [99,100]. Furthermore, the compound discussed in this work is characterized by the ability to reduce cholesterol and triglyceride levels in low-density lipoproteins in the metabolic syndrome. It was found that it also decreased the level of transaminases and improved insulin resistance [20,101,102]. It has been shown that curcumin can inhibit signal transducer and activator of transcription 3 (STAT3) activation in human adipocytes [103]. STAT3 signaling network plays pathophysiological roles in lipid and glucose metabolism and immune function. This protein is a key inflammatory pathway associated to insulin resistance in obese patients [104]; therefore, curcumin may play a key function in reducing inflammation and then insulin resistance in obese patients. These findings highlighted the potential of STAT3 as a future therapeutic target for treating of obesity with the use of curcumin.

In conclusion—numerous studies have shown that curcumin could down- or upregulate various transcription factors, enzymes, cytokines and other signaling pathway components (Table 1). 

Downregulated or inactivated molecules included SCD (Stearoyl-CoA desaturase), HSL (hormone-sensitive lipase), PPARγ, SREBP-1c, C/EBP-α, FAS, ACC, SREBP-1c, NF-κB, IL-6 and TNF- α. Those upregulated or activated included SIRT1 (sirtuin), Nrf2 (nuclear factor erythroid 2-like 2) and AMPK (Figure 3).

## 7. Stimulation of Energy Expenditure

Basal metabolic rate, thermic result of food intake and physical activity thermogenesis are the three points that induce the total daily energy expenditure [126]. Thermogenesis provides heat energy to maintain body temperature. Brown adipocytes play significant roles in regulating adiposity by releasing excess energy through non-shivering thermogenesis. This process in BAT is regulated by uncoupling protein 1 (UCP1). Uncoupling protein 1 reduces the proton gradient and uncouples oxidation from ATP synthesis. This constituent, mainly located within the mitochondria inner membrane, mediates thermogenesis in response to cold exposure, diet and other environmental factors [76].

Brown adipocytes are activated by sympathetic pathways through β3-adrenoreceptors in response to various factors, e.g., food [127]. UCP3—a protein homologous to UCP1—also facilitates thermogenesis through the regulation of thyroid hormones, leptin and β3-adrenergic agonists. Thus, regulation of energy expenditure through the activation of BAT by modulation of UCP seems to be real strategy for the control of obesity [128]. Curcumin can also enhance energy expenditure. This compound stimulates thermogenesis efficiently by interfering with AMPK, SIRT1, proliferator-activated-receptor-gamma-coactivator1-α (PGC-1 α), catechol-*O*-methyltransferase (COMT) and sympathetic nervous system (SNS) activity—all of which play important roles in transcriptional regulation and physiology of adipocytes [129].

Wang et al. [130] found that with regard to C57BL/6 mice on a regular low-fat chow diet with 50-day daily curcumin gavage (50–100 mg/kg body weight per day), their WAT showed the emergence of the beige adipocytes that are associated with elevated thermogenic gene expression and mitochondrial biogenesis. The outcome of this experimental work demonstrated that curcumin promoted β3AR gene expression in the inguinal white adipose tissue and elevated the levels of plasma norepinephrine, a hormone that can induce WAT browning [130]. Song et al. [131] revealed that in high-fat diet fed C57BL/6 mice, curcumin intake reduced WAT macrophage infiltration and altered macrophage functional polarity in white adipose tissue [131]. The intervention also increased body temperature and energy expenditure in response to cold challenge—an effect associated with increased UCP1 expression in the BAT.

It is important to mention that the stimulatory effect on UCP1 expression or WAT browning has been documented for many other dietary phytochemicals (piperine, resveratrol and quercetin) [65,132,133]. Therefore, further experiments are necessary to identify the existence of a common pathway that mediates the effect of plant second metabolites on WAT browning and BAT UCP1 expression.

## 8. Biosafety, Toxic Reaction and Curcumin–Drug Interactions

Overall, the polyphenolic compounds derived from food are considered to be safe because they are a part of the traditional human diet [134]. It is known, however, that polyphenols could also have important anti-nutritive and pro-oxidative effects at higher doses [1]. WHO Expert Committee, on assessing the outcome of two clinical trials, placed the recommended allowable daily intake (ADI) of curcumin at 0–3 mg kg^−1^ [135]. According to the European Food Safety Authority (EFSA), the maximum permitted limit ingestion of curcumin as a food additive (E 100) ranges between 20 and 500 mg kg^−1^ of food, although the EFSA report did not disagree with the WHO recommended ADI for curcumin [136]. 

Because of the poor bioavailability of curcuma, the tolerance of this compound in high single oral doses appears to be high. In the work of Lao et al. [27], single oral doses of curcumin up to 12,000 mg did not cause any major harmful effects. Moreover, oral use of turmeric and curcumin did not demonstrate reproductive toxicity in animals at elevated doses. Studies on humans also did not show toxic effects, and curcumin was found to be safe at the dose of 6 g/day orally for 4–7 weeks. Furthermore, oral bioavailable formulations of curcumin were safe for humans at a dose of 500 mg two times a day for 30 days, but still few trials have been conducted [137]; thus, more work is necessary to establish safe effective dose levels. 

Curcumin, when consumed in amounts consistent with a balanced diet, is a substance with minimal risk of side effects. Generally, if such occur, they are usually individual cases and include the side effects of taking curcumin in increased doses, including, e.g., inhibition of platelet aggregation and, as a result, anticoagulant properties, a temporary increase in the activity of liver enzymes, gastrointestinal disorders or contact dermatitis and hives [138]. Because curcumin reduces pathology in transgenic models of Alzheimer’s disease (AD), a study was conducted to, among others, generate tolerance levels of curcumin (2 g or 4 g a day for 48 weeks) in persons with AD. A total of 11.4% of all patients withdrew from the study while on curcumin, due to gastrointestinal side effects, while two patients had dark stools consistent with melena. None of the participants showed unequivocal evidence of gastrointestinal hemorrhage or hemodynamic compromise. A subgroup of people underwent bleeding time assessments and no significant effect of curcumin was observed [139]. However, a preclinical study has been conducted on the toxicity of liposomal curcumin in an animal model (dogs). The dose-dependent hemolysis occurred when the treatment dose was over 20 mg/kg [17].

Curcumin can affect xenobiotic metabolizing enzymes via the modulation of phase I metabolism, influence on phase II metabolisms and effect on membrane transporters [140]. Importantly, curcumin, when used concomitantly with conventional pharmacological drugs, may result in their pharmacokinetic changes, such as maximal plasma concentration (Cmax) and area under the concentration time curve (AUC) [141]. The mechanisms of pharmacokinetic interaction involve the inhibition of P-glycoprotein and cytochrome P450 (CYP) isoenzymes [141,142]. Drugs that interact with curcumin include antidepressants, anti-coagulants, antibiotics, chemotherapeutic agents and antihistamines [141]. An important issue is the interaction between curcumin, and drugs used in the treatment of cardiovascular disorders. Curcuminoids (including curcumin) affect cytochrome P450 CYP1A2 (approximately 13% of all CYP enzymes in the human liver) and are responsible for the metabolism of arylamines (e.g., theophylline). Another example is the inhibition of CYP2C19 by curcuminoid extract. This cytochrome is responsible for the metabolism and conversion of clopidogrel (as a pro-drug) to its active metabolite. Curcumin is also known to increase the blood levels of losartan and its metabolite, and to cause clinical interactions with clopidogrel, leading to gastrointestinal bleeding (via CYP3A4 inhibition). Curcumin changes the pharmacokinetics of many cardiovascular drugs—such as rosuvastatin (by inhibiting organic anion transporting polypeptides transporters) or warfarin and clopidogrel (inhibiting P-glycoprotein in plasma) [143]. Furthermore, curcumin has a synergistic effect on inflammatory pain management with diclofenac and synergistic anticancer effect in colon cancer with silymarin. The use of the combination of the extract and the drug resulted in a reduction the tumor growth and showed beneficial efficacy in mice bearing colon cancer [142]. It has been noted that curcumin treatment potentiates the cytotoxic effect of temozolomide and etoposide in U-87NG and D283 brain tumor cells. It was caused by increasing Bax/Bcl-2 ratio and downregulating the mRNA expression of p10 and p53 [144]. 

## 9. Summary and Conclusions

There is an increasing awareness of the health benefits of naturally derived plant polyphenols. An insightful body of knowledge on polyphenols is available, covering the properties and mechanisms of action of natural polyphenols that underpin the prevention and management of obesity. 

Oxidative stress plays a significant role in the pathophysiology of obesity. It does so by modifying the concentration of molecules taking part in inflammation (which is associated with a large number of adipocytes), by promoting lipogenesis and stimulating the differentiation of preadipocytes to mature adipocytes adipogenesis and by altering the function of regulatory factors of mitochondrial activity. Therefore, the natural plant antioxidants can be important in controlling obesity. 

Unfortunately, understanding the role of polyphenols encounters a lot of problems, due to their multiple targets and the absence of specific receptors. Therefore, mechanisms underlying the triggering by curcumin of many signaling cascades in special manners, are far from transparent. Another problem, difficult to overcome, is low absorption of this compound and its interactions with the gut microbiota. An important issue regarding turmeric is the herb–drug interactions that can occur in treatments for various diseases. It is expected that rigorous studies, including human trials, will provide more information on the effective use of curcumin in preventing and managing obesity in humans. 

In summary, curcumin is a promising natural bioactive compound which possesses numerous pharmacological activities. It shows multiple mechanism of action and can affect cellular biochemical and physiological regulation. In pharmacies, there are more and more preparations containing curcumin recommended for the prevention of obesity. However, more comprehensive and definitive human studies should be performed to assess the possibility of using curcumin in preventing and managing obesity in humans.

## Figures and Tables

**Figure 1 ijms-23-00639-f001:**
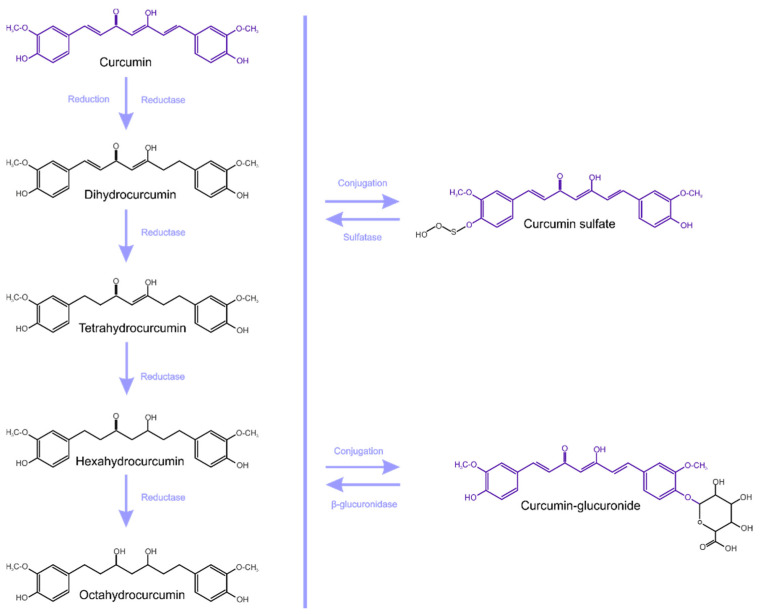
Metabolism of curcumin (conjugation and reduction).

**Figure 2 ijms-23-00639-f002:**
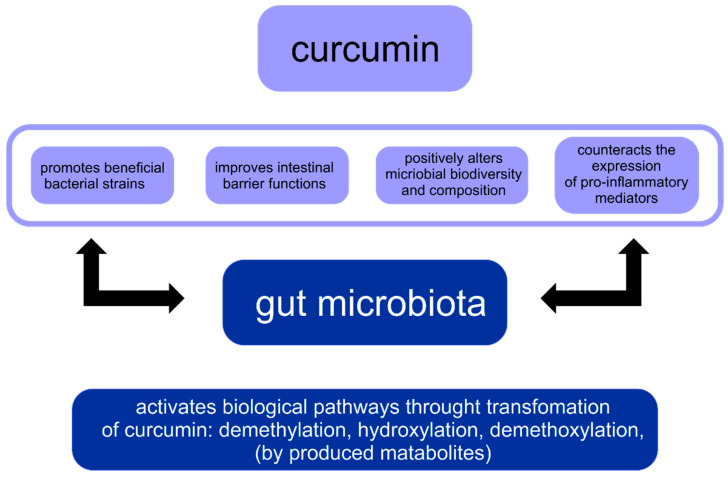
Reciprocal interaction between curcumin and gut microbiota.

**Figure 3 ijms-23-00639-f003:**
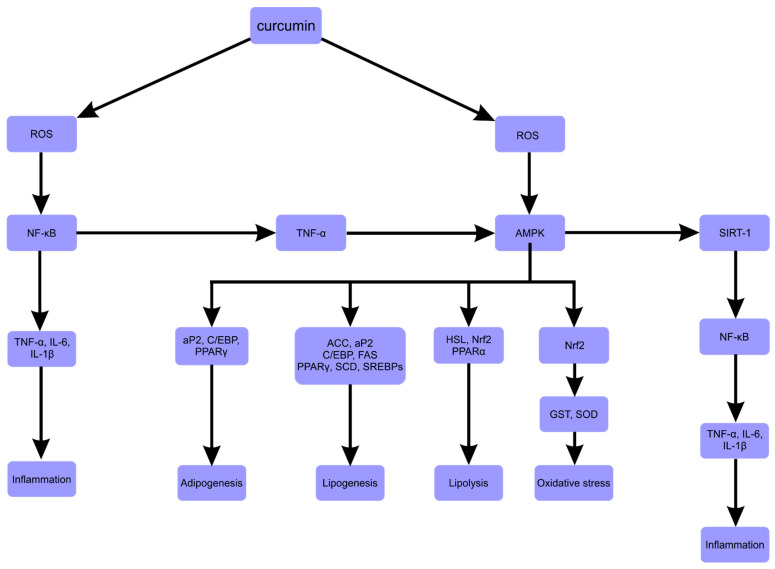
Potential anti-obesity mechanisms of curcumin [125]. An explanation of the abbreviations can be found at the end of the review. Adapted with permission from ref. [125]. Copyright 2021 Ohishi, Fukutomi, Shoji, Goto and Lsemura.

**Table 1 ijms-23-00639-t001:** Effect of curcumin on obesity in human, animal and in vitro studies. An explanation of the abbreviations can be found at the end of the review.

Type of Study	Experimental Design and Treatments	Results	Trial Lenght	Ref.
Human study	Randomized, controlled trial; 44 obese subjects supplemented with curcumin complexed with phosphatidylserine in phytosome form	↑BW loss ↑Enhance body fat reduction ↑Waistline reduction ↑Hip circumference reduction ↓BMI	30 days	[105]
Human study	Randomized, double-blind, crossover trial; 30 obese individuals supplemented with capsules containing 500 mg curcuminoids plus 5mg bioperine; two capsules per day	↓Serum IL1β and IL-4 levels	4 weeks	[106]
Human study	Randomized, double-blind, placebo-controlled crossover trial; 30 obese individuals supplemented with capsules containing 500 mg curcuminoids plus 5 mg bioperine; one capsule per day	↓Serum triglycerides level	30 days	[107]
Human study	Randomized, doubled-blinded, placebo-controlled, crossover design, 62 overweight/obese females with systemic inflammation supplemented with turmeric; 800 mg per day	No significant changes in none of the investigated metabolic parameters or inflammation	10 weeks	[108]
Human study	Randomized, double-blinded, placebo-controlled trial. Elderly (n = 36, N 50 years); placebo group, 1 g per day curcumin group and 4 g per day curcumin group	↔Serum lipid profile (triglycerides, total, LDL-C, HDL-C)	6 months	[109]
Human study	A pre–post study. Healthy adults (n = 8, 43–70 years) received 10 mg per day	↓Serum LDL, Apo B, Apo B/Apo A ↑Serum HDL and Apo A	30 days	[110]
Animal study	Male Sprague–Dawley rats in an HFD-induced obesity model; control group, curcumin 100 mg/kg/BW per daygroup, 400 mg/kg/BW per day, HFD group, HFD + curcumin 100 or HFD + curcumin 400	↓Liver triglYcerydes ↓Serum fetuin-A	8 weeks	[111]
Animal study	Male C57BL/6 J mice (8 weeks old) in an HFD-induced obesity and insulin resistance model; LFD group, HFD group and HFD + curcumin group—50 mg/kg BW by gavage	↔BW ↑NrF2 in skeletal muscle ↑Glucose disposal and insulin sensitivity ↓MDA and ROS in skeletal muscle and mitochondria	15 days	[112]
Animal study	Male C57BL/6 J mice (5 weeks old) in an HFD-induced obesity model; LFD group, HFD group or HFD + curcumin (4 g/kg diet, added 2 days/week) group	↓BW and fat ↓NF-κB ↓SREBP-1c ↔Wnt signaling in mature adipocytes ↓Inflammatory in adipocytes	28 weeks	[113]
Animal study	European obese cats (6.5 years old); control group, citrus group or curcumin group	↓IFN-γ and IL-2 mRNA levels ↔mRNA expression of TNF-α, IL-1β, IL-4, IL-5, IL-10, IL-12, IL-18, TGF-β	8 weeks	[114]
Animal study	Male C57BL/6 J, ob/ob mice and nonobese littermates in a model of steatosis; ob/ob control group, lipo group, ob/ob. + curcumin group, nonobese control group or nonobese + curcumin group	↓NF-κB pathway ↓TNF ↓IL-6 ↑ IL-4 ↑Insulin sensitivity ↑Serum adiponectin	24 or 72 h	[115]
Animal study	Male C57BL/6 mice (4 weeks old) in an HFD-induced obesity model; control group, HFD group or HFD + curcumin group (500 mg/kg of diet)	↓BW, fat, microvessel density in adipose tissue. ↓Liver weights and hepatic steatosis ↓Serum glucose and triglycerides ↑Fatty acid and energy metabolism (↑ P-AMPK, P-ACC mRNA expression; ↓PPARγ and C/EBPα mRNA expression)	12 weeks	[116]
Animal study	Male Wistar rats (100–120 g) in an HFD-induced obesity model; control group, HFD control group, HFD + 30 mg/kg BW curcuminoid group, HFD + 60 mg/kg BW curcuminoid group, HFD + 90 mg/kg BW curcuminoid group	↓Plasma FFA ↓glucose levels	12 weeks	[117]
Animal study	Male Golden-Syrian hamsters (4 weeks old) in an HFD-induced obesity model; HFD group or HFD + curcumin (0.05% in diet)	↔BW, food intake, fat pad mass, plasma glucose Plasma FFA, triglycerydes, leptin, insulin ↑Plasma HDL-C, Apo A-I ↓Hepatic cholesterol and triglycerydes ↑FA β-oxidation activity ↓FAS, HMG-CoA reductase ↓Lipid peroxide levels	10 weeks	[118]
Animal study	Male wild-type C57BL/6 J mice (8–10 weeks old) in a diet-induced-obesity (DIO) model. Male ob/ob C57BL/6 J mice (3–5 weeks old); groups including DIO control group, DIO + 3% curcumin, ob/ob control group or ob/ob. + 3% curcumin	↓BW and body fat ↑Glycemic status and insulin sensitivity ↓Adipose, hepatic and systemic inflammation ↑mRNA expression of adiponectin ↓Hepatic NF-κB activity	60 days	[119]
Animal study	Male Sprague–Dawley rats in an HF-diet-induced obesity model; control group, high curcumin (5.00 g/kg BW, HFD group, HFD + low curcumin (1.25 g/kg diet) group or HFD + high curcumin (5.00 g/kg diet) group	↓BW, blood glucose, insulin, leptin, TNF-α ↓Insulin resistance and leptin resistance	4 weeks	[120]
In vitro study	3T3-L1 cells treated with curcumin (0–30 μM)	↑Apoptosis at 30 μM ↓Glycerol release ↓MAPK phosphorylation	2 and 24 h	[121]
In vitro study	Primary cell culture from epididymal fat pads treated with curcumin (0–20 μM)	↔Wnt signaling ↓Inflammatory and oxidative pathway ↓NF-κB signaling	0–60 min	[113]
In vitro study	3T3-L1 cells treated with curcumin (0–100 μM)	↓ Adipocyte differentiation and lipid accumulation ↓FAS, PPARγ	0–8 days	[122]
In vitro study	Rabbit subcutaneous adipocytes treated with curcumin (0–20 μg/mL)	↑Cholesterol efflux from adipocytes ↑PPARγ, LXRα, ABCA1	24 h	[123]
In vitro study	3T3-L1 cells treated with curcumin (0–30 μM)	↓Adipocyte differentiation and fat accumulation ↓Cell viability ↓C/EBPβ, PPARγ and C/EBPα	18, 24 and 48 h, 6 days	[77]
In vitro study	3T3-L1 cells treated with curcumin (0–50 μM) for	↑AMPK PPARγ ↑Phosphorylation of ACC ↓Fat accumulation	8 days	[124]
In vitro study	3T3-L1 cells treated with curcumin (0, 5,10 and 20 μM)	↓Adipocyte differentiation, fat accumulation and adipogenesis ↑fat oxidation ↓ACC ↑AMPK activation ↑Apoptosis	24 h	[116]

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
