# Peer review of "Curcumin and Weight Loss: Does It Work?"

_ijms, 2022, doi:10.3390/ijms23020639_

Round 1
Reviewer 1 Report
In this review Kasprzak-Drozd and colleagues analysed the current literature regarding the role of curcumin as anti-obesity agent and its effectiveness in managing obesity.
This is a very interesting and well written review that highlights the current findings regarding the role of this compound in managing cell metabolism.
Only few points need to be integrated in order to highlight the role of curcumin as potential treatment and/or co-treatment of metabolic disorders.
- Chapter 2 "Medicinal activities, bioavailability and metabolism of curcumin - brief overview" in order to highligh the multifunction role in non-cancerous diseases it should be added that curcumin plays also a key role il pregnancy outcome (see PMID: 33477354)
- Chapter 6 "Regulation of lipid metabolism". It should be added that since it has been shown that curcumin can inhibit STAT3 activation in human adipocytes (see PMID: 31781039), a key inflammatory pathway associated to insulin resistance in obese patients (see PMID: 29635003), curcumin may play a key function in reducing inflammation and then insulin resistance in obese patients. This may also further explain the increased Insulin sensitivity found in the clinical trials mentioned by the authors.
- an accurate revision of typing errors is recommended
Author Response
The authors would like to thank the Reviewer for valuable comments which have helped to improve the quality of the manuscript. We hope that the revisions in the manuscript and our accompanying responses will be sufficient to make our manuscript suitable for publication.
In this review Kasprzak-Drozd and colleagues analysed the current literature regarding the role of curcumin as anti-obesity agent and its effectiveness in managing obesity. This is a very interesting and well written review that highlights the current findings regarding the role of this compound in managing cell metabolism. Only few points need to be integrated in order to highlight the role of curcumin as potential treatment and/or co-treatment of metabolic disorders.
- Chapter 2 "Medicinal activities, bioavailability and metabolism of curcumin - brief overview" in order to highligh the multifunction role in non-cancerous diseases it should be added that curcumin plays also a key role in pregnancy outcome (see PMID: 33477354)
Chapter 2 "Medicinal activities, bioavailability and metabolism of curcumin - brief overview" has been supplemented according to Reviewer recommendation (highlighted).
- Chapter 6 "Regulation of lipid metabolism". It should be added that since it has been shown that curcumin can inhibit STAT3 activation in human adipocytes (see PMID: 31781039), a key inflammatory pathway associated to insulin resistance in obese patients (see PMID: 29635003), curcumin may play a key function in reducing inflammation and then insulin resistance in obese patients. This may also further explain the increased Insulin sensitivity found in the clinical trials mentioned by the authors.
Chapter 6 "Regulation of lipid metabolism" has been supplemented according to Reviewer recommendation (highlighted).
- An accurate revision of typing errors is recommended.
The manuscript has been improved towards grammar and stylistics by native speaker Jack Stanley Dunster from Canada (Language Editor of Current Issues in Pharmacy and Medical Sciences), who has many years of experience in this type of work.
Reviewer 2 Report
The current paper is a narrative review that attempts to assess the effect of curcumin on weight loss. The aim seems challenging, however important methodological limitations obstacle the drawing of any firm conclusion. In quality of expert involved in the research and treatment of patient with obesity, the publication of such reports may create false expectations, especially since conclusions are obtained by means of narrative and not systematic review, where the former are highly biased.
Author Response
The current paper is a narrative review that attempts to assess the effect of curcumin on weight loss. The aim seems challenging, however important methodological limitations obstacle the drawing of any firm conclusion. In quality of expert involved in the research and treatment of patient with obesity, the publication of such reports may create false expectations, especially since conclusions are obtained by means of narrative and not systematic review, where the former are highly biased.
The authors would like to thank the Reviewer for valuable comments which have helped to improve the quality of the manuscript.
The authors strongly agree with the Reviewer that narrative review may seem biased in many cases. For this reason, we have highlighted in the Introduction and in the Conclusions that curcumin possess subtle effects through multiple mechanisms and biochemical targets collectively leading to substantial health benefits. This compound is now used as an aid in weight loss. However, more comprehensive and definitive human studies should be performed to assess the possibility of using curcumin in preventing and managing obesity in humans.
Reviewer 3 Report
It is an interesting review paper on curcumin anti-obesity mechanisms and effectiveness of curcumin for obesity management. Below are comments and suggestions.
Minor comments
In section 2. Medical activities... in sentence line 101 „Moreover, curcumin has shown anti-lipidemic effect“ add a reference(s) (for example 98 i 99) and then adjust the ones that follow.
In section 6. Regulation of lipid metabolism line 443, there is a double space between the words intervention and in animals.
The authors use abbreviations and full names in the table 1. for the timestamps. For the hour mark they use the abbreviation h (commonly used abbreviation is hrs), while for others full name (days, weeks, months), so this has to be corrected. In addition, in table 1. in the column named Type of study, the word study is misspelled. In the same table in the column named Trail length, there is a period (dot) after some timestamps (day, weeks, h), so this has to be corrected too.
Major comments
In section 8. Biosafety and toxic reaction 1) comment on herb–drug pharmacokinetic interactions - conventional drugs in concomitant use with curcuminoids 2) rename the section title to Biosafety ,toxic reaction and curcumin - drug interactions
In Conclusion also comment on herb–drug interactions.
Author Response
The authors would like to thank the Reviewer for valuable comments which have helped to improve the quality of the manuscript. We hope that the revisions in the manuscript and our accompanying responses will be sufficient to make our manuscript suitable for publication.
It is an interesting review paper on curcumin anti-obesity mechanisms and effectiveness of curcumin for obesity management. Below are comments and suggestions.
Minor comments
- In section 2. Medical activities... in sentence line 101 „Moreover, curcumin has shown anti-lipidemic effect“ add a reference(s) (for example 98 i 99) and then adjust the ones that follow.
It has been corrected.
- In section 6. Regulation of lipid metabolism line 443, there is a double space between the words intervention and in animals.
A double space has been removed.
- The authors use abbreviations and full names in the table 1. for the timestamps. For the hour mark they use the abbreviation h (commonly used abbreviation is hrs), while for others full name (days, weeks, months), so this has to be corrected. In addition, in table 1. in the column named Type of study, the word study is misspelled. In the same table in the column named Trail length, there is a period (dot) after some timestamps (day, weeks, h), so this has to be corrected too.
Errors and typos in the table have been corrected according to Reviewer comments.
Major comments
- In section 8. Biosafety and toxic reaction 1) comment on herb–drug pharmacokinetic interactions - conventional drugs in concomitant use with curcuminoids 2) rename the section title to Biosafety ,toxic reaction and curcumin - drug interactions. In Conclusion also comment on herb–drug interactions.
Thank you for your comment allowing to complete the data contained in the article. The title of the section has been changed and the curcumin - drug interactions subject has been discussed in the text of the section and commented in the Conclusion.
Round 2
Reviewer 2 Report
Thank you.
Reviewer 3 Report
The authors have dealt with my criticisms adequately.
Minor comment
In section 2. Medicinal activities, bioavailability and metabolism of curcumin - brief overview line 137, there is a large space between the words 1,7-bis(4-hydroxy-3-methoxyphenyl)hept-1-ene-3,5-dione and 1,7-bis(4-hydroxy-...